# The Efficacy and Safety of Herbal Medicine with Pirfenidone in the Treatment of Idiopathic Pulmonary Fibrosis: A Systematic Review

**Suyeon Cho** [1,†] **, Sunju Park** [2,†] **, Ju Ah Lee** [3] **, Hee-Jae Jung** [4,5] **, Kwan-Il Kim** [4,5,*] **and Beom-Joon Lee** [4,5,*]

1. College of Korean Medicine, Daejeon University, Daejeon 34520, Republic of Korea
2. Department of Preventive Medicine, College of Korean Medicine, Daejeon University, Daejeon 34520, Republic of Korea
3. Hwa-pyeong Institute of Integrative Medicine, Incheon 21949, Republic of Korea
4. Department of Clinical Korean Medicine, Graduate School, Kyung Hee University, Seoul 02447, Republic of Korea
5. Division of Allergy, Immune and Respiratory System, Department of Internal Medicine, College of Korean Medicine Kyung Hee University, Kyung Hee University Medical Center, Seoul 02447, Republic of Korea
\* Correspondence: myhappy78@naver.com (K.-I.K.); franchisjun@naver.com (B.-J.L.)
† These authors contributed equally to this work.

**Abstract:** Although there were randomized control trials (RCTs) that showed the considerable efficacy of herbal medicine (HM) in idiopathic pulmonary fibrosis (IPF) and systematic reviews on the value of some herbs in the treatment of IPF, there have been no systematic reviews comparing the combined use of complex HM against pirfenidone monotherapy in IPF. This review evaluated the efficacy of parallel administration of HM and pirfenidone in IPF treatment. We conducted a systematic review of RCTs that compared pirfenidone monotherapy against pirfenidone combined administration with HM in IPF. We searched the EMBASE, CENTRAL, PubMed, and CNKI databases for relevant RCTs published before July 2021. Six RCTs were eligible for inclusion. Compared with the control group, a greater recovery or a smaller reduction in forced vital capacity (FVC) and, in general, a valid improvement in the St. George's Respiratory Questionnaire was observed in the treatment group. However, it should be noted that the risk of bias of the included RCTs was high or unclear in most categories. In IPF treatment, HM administered with pirfenidone effectively protected pulmonary function and improved the quality of life. However, given the number and quality of the included studies, the evidence was not strong enough to draw definitive conclusions. Well-designed future RCTs are warranted to evaluate the impact of HM on IPF.

**Keywords:** IPF; herbal medicine; pirfenidone; effectiveness; safety





## 1. Introduction

Idiopathic pulmonary fibrosis (IPF) is a chronic, progressive, irreversible respiratory disease characterized by scar tissue within the lungs [1,2]. It accounts for the highest proportion of idiopathic interstitial pneumonia [3]. The major symptoms are dry cough and dyspnea on exertion, often accompanied by cyanosis or clubbed fingers [4]. Smoking, aging, and exposure to elements, such as livestock or metal dust, are proven risk factors. Males are also known to be more vulnerable to IPF; however, its pathogenesis is not fully understood [5].

The prognosis of IPF is commonly known to be poor. Although the reported fatality differs, median survival since diagnosis ranges from 2.5 to 3.5 years [6,7]. The prevalence of IPF has also not been wholly identified; Maher et al. have recently proposed the prevalence to be 0.33–4.51 per 10,000 people [8]. Although the global prevalence is rising, treatment available for IPF remains limited [5,7].

As a cure for IPF has not yet been identified, it is difficult to expect a complete recovery. Due to the irreversibility of the disease, treatment aims primarily to slow the progression of the disease rather than to recover [1,2]. Currently, pirfenidone and nintedanib are the only drugs approved by the FDA for the management of IPF. According to the ATS/ERS/JRS/ALAT guidelines, both drugs received conditional recommendations for use with moderate confidence in the effect estimates [9]. However, pirfenidone and nintedanib have evident limitations. They can slow the progress of IPF and reduce the decline in forced vital capacity (FVC) [10–13], but their impact on the quality of life (QoL) or long-term mortality is debatable. There is a lack of evidence that supports these medications enhance the QoL in patients with IPF [11,13]. Moreover, the long-term effect on mortality remains unknown despite some studies reporting that pirfenidone could reduce early deaths [13,14]. Pirfenidone can also cause adverse events (AEs) such as gastrointestinal symptoms, rash, photosensitivity reaction, weight loss, headache, and arthralgia [15]. These often lead to discontinuance or modification of the treatment plan [16].

In the absence of an effective treatment for IPF, it is worthwhile to examine the safety and efficacy of complementary medicine. A systematic review by Zhang et al. [17] showed the effectiveness and safety of *Radix Astragali* and *Radix Angelicae Sinensis* in treating IPF. Several randomized clinical trials (RCTs) have already been conducted regarding the treatment of IPF with herbal medicine (HM) [18–23]. However, there has not yet been a systematic review that examined the efficacy of complex herbal medicine rather than single herbs in the treatment of IPF.

This review aims to compare the effectiveness and safety of the combined use of HM and pirfenidone with pirfenidone monotherapy, focusing on their impact on pulmonary function and QoL of patients with IPF.

## 2. Materials and Methods

The protocol of this study was pre-registered with OSF registries (https://osf.io/uqvf4; accessed on 7 July 2021). A systematic review was conducted following the Preferred Reporting Items for Systematic Reviews and Meta-Analyses (PRISMA) 2020 statement (Table S1) [24]. Ethical approval was not required, as all data and information used in this study were collected from previously published clinical trials.

### 2.1. Search Strategy and Selection Criteria

2.1.1. Search Database

The following four databases were searched: EMBASE, Cochrane Central Register of Controlled Trials (CENTRAL), PubMed, and one Chinese database (CNKI).

2.1.2. Search Strategy

We employed both medical subject headings (MeSH) and refined vocabulary as needed. The search strategies were reviewed by experts. The strategies were revised until July 2021, and no modifications were made after the final organization. Please refer to the Table S2 for the detailed search strategies.

### 2.2. Eligibility Criteria for Articles of Inclusion

RCTs comparing the use of pirfenidone alone and in combination with HM for IPF treatment were eligible for inclusion. There were no limitations in terms of sex, age, or disease severity. However, trials that did not exclude patients with other critical pulmonary or chronic systemic diseases, such as cardiovascular, hepatic, or renal disorders, were excluded.

There were no regulations on the prescription, dose, or formulation of HM administered to the treatment group. Pirfenidone was administered to all participants, but the details, such as dose, varied from study to study.

The primary measurement used in this review to evaluate pulmonary function was FVC. As secondary measures, all reported indicators in each trial were accepted and examined.

Finally, only studies published in Chinese or English were included. Non-RCT studies or trials in which the full text was not accessible were also excluded.

### 2.3. Data Extraction and Assessment

#### 2.3.1. Selection of Studies

Two authors (JAL and SP) independently examined the titles and abstracts. Studies that did not meet the inclusion criteria were excluded, and the rationale for each exclusion was documented in EndNote X9. In addition, screening for duplicate trials were performed. Any disagreements were resolved through discussions between the authors. The study selection process is summarized in a PRISMA-compliant flowchart (http://www.prisma-statement.org; accessed on 1 May 2022).

#### 2.3.2. Data Extraction

One review author (SC) extracted data from the full text of selected studies, covering the baseline information about: (1) study design; (2) risk of bias; (3) participants (e.g., number registered, number analyzed, age, major symptoms, and how long the patient was ill); (4) outcome measurements; (5) interventions (e.g., prescription, composition, dose, treatment period, number of treatment sessions, subsidiary treatment); and (6) the results of each trial. After extraction, the data were abstracted into a table. Another review author (SP) reexamined the extracted data.

#### 2.3.3. Risk of Bias Assessment

One review author (SC) assessed the risk of bias (RoB) of the included trials based on the Cochrane Handbook for Systematic Reviews of Interventions version 5.1.0. [25]. Another review author (SP) double-checked the accuracy afterward.

We used the Cochrane Collaboration's RoB tool, which covers the following elements: random sequence generation, allocation concealment, blinding of participants and personnel, blinding of outcome assessment, incomplete outcome data, selective reporting, and other sources of bias [25]. Evaluation results were presented with abbreviations: "L" for low RoB, "U" for unclear risk, and "H" for high risk.

#### 2.3.4. Data Synthesis

The differences between the intervention and control groups were assessed. We used mean differences (MDs) with 95% confidence intervals (CIs) for continuous data to measure treatment effects. We converted other forms of data into MDs. For outcome variables with different scales, we used the standardized mean difference with 95% CIs. For dichotomous data, we presented treatment effects as relative risk with 95% CIs. All statistical analyses were performed with the Cochrane Collaboration's software program Review Manager (RevMan) version 5.2.7 (Copenhagen, The Nordic Cochrane Center, the Cochrane Collaboration, 2012) for Windows. When appropriate, we pooled the data across studies for meta-analysis using random effects. Intention-to-treat analyses, including all randomized patients, were performed. Chi-square and I-squared tests were used to evaluate the heterogeneity of the included studies.

## 3. Results

### 3.1. Study Selection

After searching EMBASE, CENTRAL, PubMed, and CNKI, 2135 studies were extracted and, finally, six trials were included in this review. Every study included was a non-blinded, two-arm, randomized parallel controlled trial conducted and published in China (Figure 1).

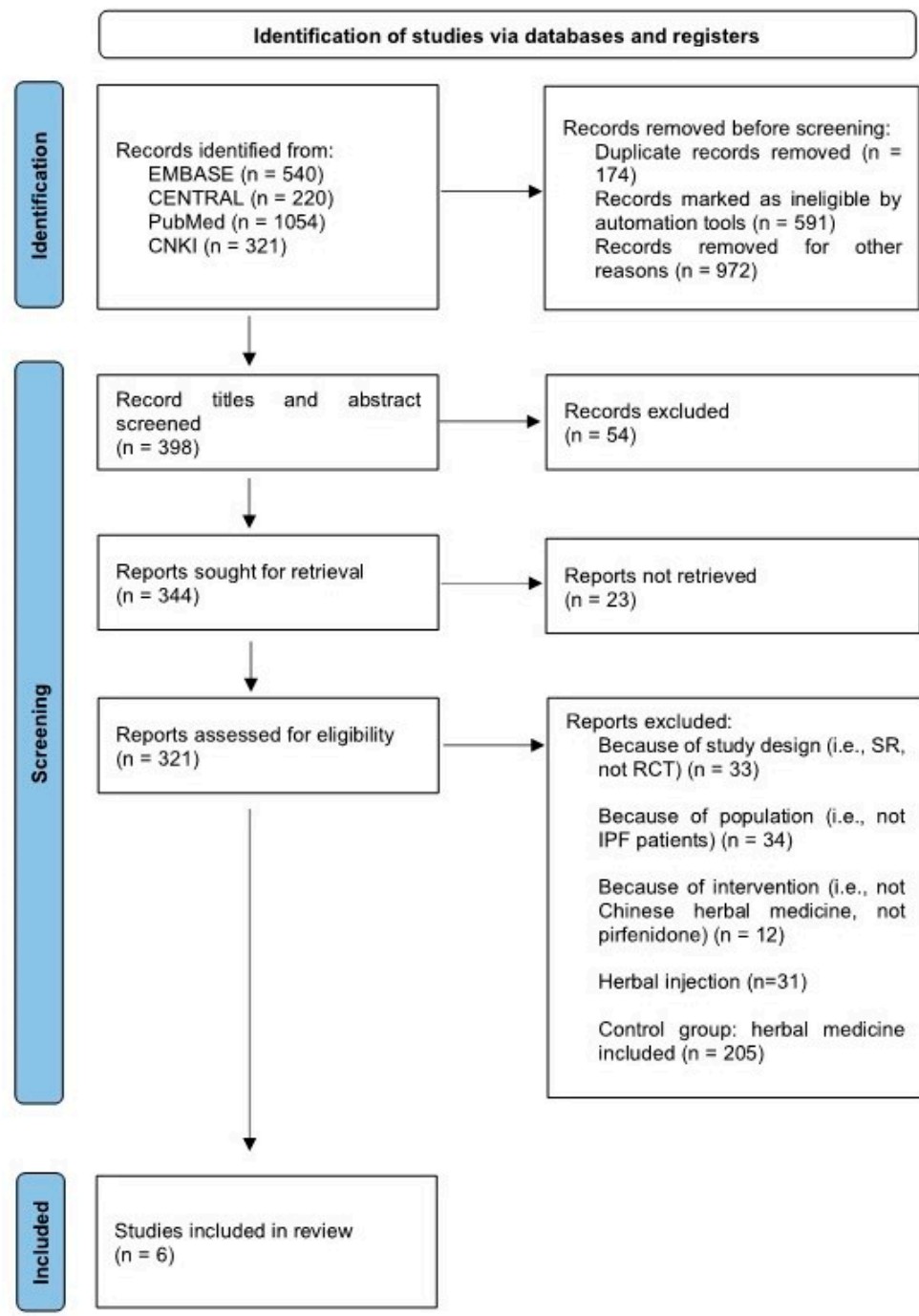

**Figure 1.** PRISMA flow diagram.

### 3.2. Summary of Findings (SoF)

#### 3.2.1. Baseline Characteristics

The six studies reviewed aimed to compare the efficacy and safety of HM with pirfenidone alone. The treatment and control groups in the six studies were patients with pure IPF. Patients with IPF and other critical diseases were excluded from the screening process. Similarly, patients with allergies were excluded except for one trial [18]. None of the studies had restrictions on IPF status, such as severity, presence of complications, or how long the patient has been ill. When identifying patients with IPF, some trials exclusively adopted Western diagnostic criteria [18,19], while others applied additional criteria from a traditional Chinese medicine (TCM) perspective [20–23].

In addition, half of the trials set criteria to rate the curative effects [18,22,23]. Indices, such as high-resolution computed tomography (HRCT) score, forced expiratory volume in 1 s (FEV1), diffusing capacity of the lung for CO (DLCO), and TCM symptom scores, were used to define valid advancements. Based on this standard, patients were divided into two groups: those with valid enhancements and those without. The improvement rate in the HM treatment group, based on this classification, ranged from 80% [23] to 94.92% [18]. Meanwhile, the control group showed a rate of 65–77.12% [18,22]. The difference was statistically significant in all studies ($p < 0.05$). Furthermore, the degree of enhancement was further specified at 2–3 levels.

Only Liu et al. [23] conducted a follow-up study. During the follow-up period, both groups progressed in all dimensions of SF-36. In both groups, the frequency of recurrence decreased, with an improvement in overall symptoms and stamina. Moreover, every improvement was more significant in the HM treatment group than in the control group.

Regarding the outcome measurements, we used two indicators as the primary measurements: FVC to evaluate pulmonary function and St. George's Respiratory Questionnaire (SGRQ) to evaluate the QoL. The SGRQ can be rated in four areas: symptom score (evaluating the severity and frequency of symptoms); activity score (evaluating the impact of dyspnea in terms of physical activity and mobility); impact score (evaluating the impact of the disease on the psychosocial aspect); and total score. The total score ranges from 0 to 100, with a lower score indicating a higher QoL [26].

Of the six trials, two presented FVC in percentage [20,22], and two in liters [19,21]. The rest did not report on FVC [18,23]. Four studies that reported FVC showed considerable advancement compared to pretreatment outcomes. Furthermore, the extent of the increase was greater than that in the control group [19–22].

Meanwhile, only one trial recorded all four dimensions of SGRQ [20]. One documented total score alone [18], and one referred to all dimensions but total score [21] and the remaining did not make reports on SGRQ [19,22,23]. Li (2018) [18] presented the total score of SGRQ, and improvement was seen in both groups. Advancement was also highly significant ($p < 0.05$) compared to the control. Although Bai (2019) [21] did not mention the total score, the scores showed uplifts in all three categories compared to the control. In Gu (2018) [20], scores for all three categories and the total score underwent considerable improvement compared to pretreatment. Among them, the impact score and total score showed notable enhancement compared to the control.

There were no restrictions on secondary outcome measures, and the indicators covered were as follows: (1) symptomatic indicators such as symptom score, dyspnea score, TCM symptom score, and modified medical research council dyspnea scale (mMRC) [18,20–23]; (2) AEs, which were mainly gastrointestinal symptoms such as nausea and skin-related symptoms such as rashes [19,22,23]; (3) SF-36, which also reflects the QoL [23]; (4) other indicators that reflect pulmonary function, such as FEV1, FEV1/FVC, DLCO, MMEF, and PaO2 [18–23]; (5) HRCT score [20]; (6) serum cytokine levels reflecting inflammation levels, in which higher concentrations indicate more inflammation [19]; (7) pulmonary fibrosis index such as serum HA, LN, and PCIII [21]. Refer to Table 1 for detailed information about each trial included in the review.

### 3.2.2. Intervention

All studies reported the composition of each HM, clarifying the extent to which each herb was used in the decoctions. The six studies applied pirfenidone to the control and HM treatment groups. The treatment period ranged from 4 [18] to 36 weeks [19], and patients in both groups took their medication two [18–20,22,23] or three times a day [21].

For the control group, pirfenidone was the only major intervention. Minor interventions that would not affect test results, such as symptomatic treatment or antibiotics, were permitted. Prednisone, a glucocorticoid, was administered along with pirfenidone in a single trial [23].

**Table 1.** Basic information of included studies (N = 6).

| Author (Year) | Study Design | Sample Size (Enrolled) (I:C) | Population | | | Intervention | Control | I & C Total Tx Period/No. of Sessions | Outcome & Results I(n):Mean ± SD, C(n):Mean ± SD | Adverse Events |
|---|---|---|---|---|---|---|---|---|---|---|
| | | | Age I (Mean ± SD): C (Mean ± SD) | Course of Disease * | Major Symptoms | | | | | |
| Gu (2018) [20] | Non-blinded RCT | 60 (30:30) | I: 62.3 ± 7.6, C: 63.8 ± 8.5 | I:3.2 ± 1.1 y C:3.5 ± 1.3 y | Short breath, wheezing, fatigue, weak cough | (1) Qizhu Feixian Decoction(芪术肺纤汤)1 dose(剂)/d, 200 mL/each, 2 times/d  (2) Pirfenidone 200 mg/each, 3 times/d (Maintenance dose 600 mg)  (3) Antibiotics | (1) Pirfenidone 200 mg/each, 3 times/d (Maintenance dose 600 mg)  (2) Antibiotics | 6 m/ 2 sessions  3 m for one session | 1. FVC (%) I(30):74.5 ± 10.2 [bc], C(30):67.5 ± 9.3 [b]  2. FEV1 (%) I(30):91.2 ± 7.3 [b], C(30):90.8 ± 10.6 [b]  3. FEV1/FVC (%) I(30):73.5 ± 10.5, C(30):73.4 ± 10.5  4. HRCT I(30):24.6 ± 3.8, C(30):25.1 ± 3.9  5. Symp: Short breath I(30):4.5 ± 3.3 [ac], C(30):6.7 ± 1.4 [a]  6. Symp: Weak cough I(30):2.7 ± 0.8 [ac], C(30):3.2 ± 2.1 [a]  7. Symp: Fatigue I(30):2.0 ± 0.8 [ac], C(30):2.6 ± 1.3 [a]  8. Symp: Tongue I(30):1.9 ± 0.8 [a], C(30):2.0 ± 0.8 [a]  9. SGRQ: Symptoms I(30):30.5 ± 18.4 [a], C(30):35.8 ± 25.3 [a]  10. SGRQ: Activity I(30):32.5 ± 21.6 [a], C(30):34.1 ± 20.1  11. SGRQ: Impact I(30):15.4 ± 8.9 [ac], C(30):22.5 ± 16.8  12. SGRQ: Total I(30):20.2 ± 10.2 [ac], C(30):27.6 ± 22.4 | n.r |

**Table 1.** *Cont.*

| Author (Year) | Study Design | Sample Size (Enrolled) (I:C) | Age I (Mean ± SD): C (Mean ± SD) | Course of Disease * | Major Symptoms | Intervention | Control | I & C Total Tx Period/No. of Sessions | Outcome & Results I(n):Mean ± SD, C(n):Mean ± SD | Adverse Events |
|---|---|---|---|---|---|---|---|---|---|---|
| | | | | | Population | | | | | |
| Li (2018) [18] | Non-blinded RCT | 235 (118:117) | I: 62 ± 12.56 C: 64 ± 13.47 | I: 2.5 ± 1.6 y C: 3.1 ± 1.8 y | Cough, dyspnea | (1) Zhike Huatan Decoction(止咳化痰汤) 1 dose(剂)/d, 2 times/d (2) Pirfenidone 200 mg/each, 2 times/d (3) Symptomatic treatments | (1) Pirfenidone 200 mg/each, 2 times/d (2) Symptomatic treatments | 4 w/ 4 sessions 1 w for one session | 1. FEV1 (L) I(118):1.58 ± 0.13 [ac], C(117):1.49 ± 0.16 [a] 2. DLCO [ml/(min·mmHg)] I(118):13.31 ± 4.18 [†‡], C(117):13.14 ± 3.98 [†] 3. SGRQ I(118):12.36 ± 6.92 [ac], C(117):20.31 ± 2.13 [a] 4. Dyspnea score I(118):1.31 ± 0.63 [ac], C(117):2.29 ± 0.21 [a] 5. 6 MWT I(118): n.r, C(117): n.r | None |
| Li (2020) [19] | Non-blinded RCT | 110 (55:55) | I: 58.71 ± 6.25 C: 58.85 ± 6.62 | I: 25.88 ± 0.35 m C: 25.47 ± 0.28 m | Cough, fever, dyspnea | (1) Zhike Huatan Decoction(止咳化痰汤) 1 dose(剂)/d, 250 mL/each, 2 times/d (2) Pirfenidone 400 mg/each, 3 times/d (3) Symptomatic treatments (4) Antibiotics | (1) Pirfenidone 400 mg/each, 3 times/d (2) Symptomatic treatments (3) Antibiotics | n.r/ n.r 36 w for one session | 1. FEV1 (L) I(55):2.09 ± 0.18 [bc], C(55):2.58 ± 0.19 [b] 2. FVC (L) I(55):2.56 ± 0.36 [bc], C(55):3.17 ± 0.54 [b] 3. DLCO I(55):160.28 ± 12.74 [bc], C(55):167.29 ± 13.57 [b] 4. MMEF (L/s) I(55):1.02 ± 0.09 [bc], C(55):1.38 ± 0.07 [b] 5. Serum IL-6 (pg/mL) I(55):15.97 ± 1.57 [ac], C(55):13.58 ± 1.69 [a] 6. Serum IL-12 (pg/mL) I(55):4.57 ± 1.16 [ac], C(55):6.17 ± 1.35 [a] 7. Serum IL-18 (pg/mL) I(55):10.77 ± 1.26 [ac], C(55):9.24 ± 1.03 [a] | Nausea I:1, C:4 Vomiting I:3, C:2 Anorexia I:1, C:6 Rash I:1, C:2 |

**Table 1.** *Cont.*

| Author (Year) | Study Design | Sample Size (Enrolled) (I:C) | Population Age I (Mean ± SD): C (Mean ± SD) | Population Course of Disease * | Population Major Symptoms | Intervention | Control | I & C Total Tx Period/No. of Sessions | Outcome & Results I(n):Mean ± SD, C(n):Mean ± SD | Adverse Events |
|---|---|---|---|---|---|---|---|---|---|---|
| Bai (2019) [21] | Non-blinded RCT | 84 (42:42) | I: 56.29 ± 6.53 C: 56.48 ± 6.71 | I: 2.97 ± 0.82 y C: 2.96 ± 0.84 y | Cough, short breath, fatigue, phlegm, dry mouth and throat, feverish sensation deep in the palms, soles, and chest | (1) Maimendong Decoction(麦门冬汤) 300 mL/each, 3 times/d (2) Pirfenidone 200 mg/each, 3 times/d | (1) Pirfenidone 200 mg/each, 3 times/d (2) Symptomatic treatments | 3 m/ n.r | 1. TCM symptom scores I(42):4.62 ± 1.48 [ac], C(42):8.31 ± 2.05 [a] 2. FVC (L) I(42):3.81 ± 0.85 [ac], C(42):3.07 ± 0.71 [a] 3. FEV1/FVC (%) I(42):67.46 ± 8.27 [ac], C(42):59.74 ± 7.95 [a] 4. DLCO [ml/(min·mmHg)] I(42):18.01 ± 4.65 [ac], C(42):15.73 ± 3.40 [a] 5. HA (μg/L) I(42):97.06 ± 17.24 [ac], C(42):118.59 ± 22.37 [a] 6. LN (μg/L) I(42):113.25 ± 20.58 [ac], C(42):127.69 ± 24.93 [a] 7. PCIII (μg/L) I(42):88.34 ± 16.86 [ac], C(42):96.74 ± 18.02 [a] 8. SGRQ: Symptoms I(42):38.26 ± 11.57 [ac], C(42):45.84 ± 13.85 [a] 9. SGRQ: Activity I(42):31.47 ± 9.63 [ac], C(42):37.24 ± 12.16 [a] 10. SGRQ: Impact I(42):22.76 ± 7.04 [ac], C(42):27.51 ± 8.33 [a] | n.r |
| Bian (2018) [22] | Non-blinded RCT | 40 (20:20) | I: 56.25 ± 9.61 C: 54.35 ± 8.84 | I: 20.65 ± 11.19 m C: 21.10 ± 12.01 m | Wheezing, short breath, cough, yellow phlegm, fatigue, sweating, bitter mouth, dry throat | (1) Feixian Decoction(肺纤汤) 200 mL/each, 2 times/d (2) Pirfenidone 3 times/d W1: 200 mg/each, W2–5: 400 mg/each, W6–8: 600 mg/each (3) Antibiotics | (1) Pirfenidone 3 times/d W1: 200 mg/each, W2–5: 400 mg/each W6–8: 600 mg/each (2) Antibiotics | 2 m/ n.r | 1. TCM symptom scores I(20):8.10 ± 2.63 [ac], C(20):11.30 ± 4.46 [a] 2. mMRC I(20):1.85 ± 0.88 [ac], C(20):2.55 ± 1.05 [a] 3. FVC (%) I(20):59.85 ± 3.72 [ac], C(20):53.70 ± 4.14 4. DLCO (%) I(20):55.25 ± 6.05 [ac], C(20):47.35 ± 8.61 5. PaO2 (mmHg) I(20):70.85 ± 4.12 [ac], C(20):64.30 ± 8.90 | Nausea and GI discomfort I:0, C:4 |

**Table 1.** *Cont.*

| Author (Year) | Study Design | Sample Size (Enrolled) (I:C) | Population | | | Intervention | Control | I & C Total Tx Period/No. of Sessions | Outcome & Results I(n):Mean ± SD, C(n):Mean ± SD | Adverse Events |
|---|---|---|---|---|---|---|---|---|---|---|
| | | | Age I (Mean ± SD): C (Mean ± SD) | Course of Disease * | Major Symptoms | | | | | |
| Liu (2017) [23] | Non-blinded RCT | 40 (20:20) | I: 54.68 ± 13.516 C: 54.49 ± 13.386 | I: 4.50 ± 0.892 y C: 4.25 ± 0.813 y | Cough, short breath, cloudy phlegm, wheezing, periodic fever, night sweats, weak pulse, etc | (1) Modified Maimen-dong(加减麦门冬汤) Decoction 1 dose(剂)/d, 2 times/d (2) Pirfenidone 3 times/day W1: 200 mg/each W2–5: 400 mg/each W6-: 600 mg/each (3) Prednisone W1–4: 0.5 mg/kg daily W5–12: 0.25 mg/kg daily W13: 0.125 mg/kg daily OR 0.25 mg/kg every other day (4) Symptomatic treatments (5) Antibiotics | (1) Pirfenidone 3 times/d W1: 200 mg/each W2–5: 400 mg/each W6-: 600 mg/each (2) Prednisone W1–4: 0.5 mg/kg daily W5–12: 0.25 mg/kg daily W13: 0.125 mg/kg daily OR 0.25 mg/kg every other day (3) Symptomatic treatments (4) Antibiotics | n.r/ n.r 2 m for one session | 1. TCM symptom scores I(20):5.75 ± 1.552 [ac], C(20):8.25 ± 2.016 [a] 2. FEV1 (%) I(20):61.60 ± 7.605 [ac], C(20):56.45 ± 7.681 3. DLCO [ml/(min·mmHg)] I(20):13.69 ± 2.796 [ac], C(20):11.92 ± 2.786 4. SF-36: PF I(20):45.58 ± 8.796 [ac], C(20):36.43 ± 7.989 [a] 5. SF-36: RP I(20):60.92 ± 12.786 [ac], C(20):51.83 ± 14.580 [a] 6. SF-36: BP I(20):78.11 ± 19.693 [ac], C(20):62.56 ± 18.189 [a] 7. SF-36: MH I(20):82.57 ± 14.510 [ac], C(20):70.17 ± 13.793 [a] 8. SF-36: SF I(20):55.16 ± 13.295 [ac], C(20):44.27 ± 12.486 [a] 9. SF-36: VT I(20):59.85 ± 15.593 [ac], C(20):47.57 ± 13.173 [a] 10. SF-36: RE I(20):46.58 ± 13.259 [ac], C(20):36.79 ± 15.479 [a] 11. SF-36: GH I(20):47.62 ± 12.274 [ac], C(20):35.67 ± 10.942 [a] | FOBT positive I:0, C:3 |

n.r., not reported; Tx, treatment. a, There is a statistically significant difference compared to before treatment, with improvement, $p < 0.05$, b, There is a statistically significant difference compared to before treatment, with exacerbation, $p < 0.05$, c, There is a statistically significant difference compared to the control group, $p < 0.05$, †, No statistical differences between pretreatment and posttreatment were reported, $p$ = n.r. ‡, No statistical differences between the groups were reported, $p$ = n.r, Course of disease: the period of illness, the term indicating how long the patient has been ill. dose: Unit of counting medicine in TCM. Serum IL-6, IL-12, IL-18: serum cytokine levels; higher levels indicate increased inflammation. HA, LN, PCIII: serum markers reflecting the degree of pulmonary fibrosis; higher levels indicate more severe fibrosis. y, year; m, month; w, week; d, day. Symp, symptom scores; FVC, forced vital capacity; FEV1, forced expiratory volume in 1 s; HRCT, high-resolution computed tomography; SGRQ, St. George's Respiratory Questionnaire; DLCO, diffusing capacity of carbon monoxide; 6 MWT, 6-min walk test; MMEF, mean maximal expiratory flow; HA, serum hyaluronic acid; LN, serum laminin; PCIII, serum type III procollagen; mMRC, modified medical research council dyspnea scale; PaO2, partial pressure of arterial oxygen; GI, gastrointestinal; SF-36, SF-36® Health Survey, PF, physical functioning; RP, role limitations due to physical health problems; BP, bodily pain; MH, mental health; SF, social functioning; VT, vitality, RE, role limitations due to emotional health problems; GH, general health. * Notes: The word 'phlegm' in this table indicates the TCM concept of phlegm (痰).

Although there were no rules regarding the selection of HM, each trial included a single prescription in decoction form. Some trials had the flexibility to add a few more herbs based on the symptoms of each patient. Three studies allowed some modifications [18–20], and the other three strictly applied uniform herbal medicine [21–23]. The exact composition and modification of each prescription were available. Refer to Table 2 for detailed information.

There were cases in which the prescription name of HM used in different trials was the same. The Zhike Huatan decoction was used in two trials: Li (2018) [18] and Li (2020) [19]. Moreover, Maimendong decoction, modified or not, was applied in two trials: Bai (2019) [21] and Liu (2017) [23], but the composition was not completely identical.

**Table 2.** The composition of the herbal medicine (N = 6).

| Prescription | Composition |
|---|---|
| Gu (2018) [20]<br>Qizhu Feixian Decoction<br>(芪术肺纤汤) | *Astragali Radix(黄芪) 30–100 g, Curcumae Rhizoma(莪术) 15 g, Sparganii Rhizoma(三棱) 15 g, Polygonati Rhizoma(黄精) 15 g, Morindae officinalis Radix (巴戟天) 15 g, Pruni Semen(苦杏仁) 10 g, Scorpio(全蝎) 4 g*<br>*\*Additional herbs for modification: Panax quinquefolius L.(西洋参) 15 g, Ophiopogonis Radix(麦冬) 15 g, Lilii Bulbus(百合) 15 g, Rehmanniae Radix(生地黄) 12 g, Rehmanniae Radix preparata (熟地黄) 12 g, Schisandrae Fructus (五味子) 8 g; Fossilia Ossis Mastodi preparata(煅龙骨) 30 g, Ostreae Concha preparata(煅牡蛎) 30 g, Euryales Semen(芡实) 15 g, Psoraleae Fructus(补骨脂) 15 g, Epimedii Herba(淫羊藿) 10 g; Poria Sclerotium (茯苓) 15 g, Citri Unshius Pericarpium(陈皮) 12 g, Pinelliae Rhizoma preparata(法半夏) 10 g, Bambusae Caulis in taeniam(竹茹) 10 g* |
| Li (2018) [18]<br>Zhike Huatan Decoction<br>(止咳化痰汤) | *Houttuyniae Herba(鱼腥草) 30 g, Poria Sclerotium(茯苓) 15 g, Trichosanthis Semen(瓜蒌) 12 g, Peucedani Radix(前胡) 12 g, Perillae Fructus(紫苏子) 10 g, Magnoliae Cortex(厚朴) 10 g, Pinelliae Tuber(半夏) 10 g, Armeniacae Semen(杏仁) 10 g, Citri Unshius Pericarpium(陈皮) 8 g, Platycodonis Radix(桔梗) 6 g, Glycyrrhizae Radix et Rhizoma(甘草) 6 g, Fritillariae Cirrhosae Bulbus(川贝母) 5 g*<br>*\*Additional herbs for modification: Astragali Radix(黄芪), Ginseng Radix(人参); Salviae Militiorrhizae Radix(丹参), Cnidii Rhizoma(川芎); Coicis Semen preparata(炒薏苡仁), Atractylodes macrocephala Koidzumi(白术); Psoraleae Semen(补骨脂), Corni Fructus(山萸肉); Jiao Sanxian(焦三仙), Galli Gigeriae Endothelium Corneum(鸡内金)* |
| Li (2020) [19]<br>Zhike Huatan Decoction<br>(止咳化痰汤) | *Poria Sclerotium(茯苓) 15 g, Mori Radicis Cortex(桑白皮) 12 g, Scutellariae Radix(黄岑) 10 g, Gardenia jasminoides(山栀子) 10 g, Fritillariae Cirrhosae Bulbus(川贝母) 10 g, Citri Unshius Pericarpium(陈皮) 10 g, Peucedani Radix(前胡) 10 g, Platycodonis Radix(桔梗) 10 g, Liriopis seu Ophiopogonis Tuber(麦冬) 10 g, Alstonia Scholaris Folium preparata(灯台叶) 8 g, Glycyrrhizae Radix et Rhizoma(甘草) 6 g, Anemarrhenae Rhizoma(知母) 6 g*<br>*\*Additional herbs for modification: Amomi Fructus(砂仁); Perilla frutescens var. acuta KVDO(苏叶), Mori Folium(桑叶); Lilii Bulbus(百合), Scrophulariae Radix(玄参)* |
| Bai (2019) [21]<br>Maimendong Decoction<br>(麦门冬汤) | *Liriopis seu Ophiopogonis Tuber(麦冬) 35 g, Oryzae Semen(粳米) 20 g, Astragali Radix(黄芪) 20 g, Ginseng Radix(人参) 15 g, Schisandrae Fructus(五味子) 15 g, Corni Fructus(山萸肉) 15 g, Gecko(蛤蚧) 15 g, Cnidii Rhizoma(川芎) 15 g, Glycyrrhizae Radix preparata(炙甘草) 15 g, Lumburicus(地龙) 10 g, Pinelliae Tuber(半夏) 5 g, Zizyphi Fructus(大枣) 3pcs* |
| Bian (2018) [22]<br>Feixian Decoction<br>(肺纤汤) | *Poria Sclerotium(茯苓) 15 g, Codonopsis Pilosulae Radix(党参) 10 g, Liriopis seu Ophiopogonis Tuber(麦冬) 10 g, Fritillariae Cirrhosae Bulbus(川贝母) 10 g, Schisandrae Fructus(五味子) 10 g, Astragali Radix(黄芪) 10 g, Atractylodes macrocephala Koidzumi(白术) 10 g, Polygonum cuspidatum Siebold et Zuccarinii(虎杖) 10 g, Magnoliae Cortex(厚朴) 10 g, Mori Folium(桑叶) 10 g, Tangerine pith(橘络) 10 g, Cnidii Rhizoma(川芎) 10 g, Platycodonis Radix(桔梗) 10 g, Trichosanthis Semen Cortex(瓜蒌皮) 10 g, Mori Radicis Cortex(桑白皮) 10 g, Glycyrrhizae Radix et Rhizoma(甘草) 3 g* |

**Table 2.** *Cont.*

| Prescription | Composition |
| --- | --- |
| Liu (2017) [23]<br>Modified Maimendong Decoction<br>( 加减麦门冬汤) | *Liriopis seu Ophiopo gonis Tuber*(麦冬) *15 g, Codonopsis Pilosulae Radix*(党参) *15 g, Adenophora stricta Miq.*(南沙参) *15 g, Poria Sclerotium*(茯苓) *15 g, Astragali Radix*(黄芪) *15 g, Salviae Militiorrhizae Radix*( 丹参) *15 g, Pinelliae Rhizoma preparata*( 法半夏) *10 g, Atractylodes macrocephala Koidzumi*(白术) *10 g, Polygonati Odorati Rhizoma*(玉竹) *10 g, Trichosanthis Radix*(天花粉) *10 g, Poly gonum cuspidatum Siebold et Zuccarinii*(虎杖) *10 g, Cnidii Rhizoma*(川芎) *10 g, Psoraleae Semen*(补骨脂) *10 g, Glycyrrhizae Radix et Rhizoma*(甘草) *5 g* |

Gu (2018) [20] applied the Qizhu Feixian decoction, and modifications were made according to the symptoms of each patient. Both the control and HM treatment groups received antibiotics as needed. However, drugs that could affect the study results, such as hormones and immunosuppressants, were not used.

Li (2018) [18] used the Zhike Huatan decoction, and modifications were allowed. Symptomatic treatments were available to all participants when necessary (e.g., oxygen therapy to relieve hypoxemia and regulation of acid-base/electrolyte imbalance).

Li (2020) [19] applied the Zhike Huatan decoction with occasional modifications. Before taking pirfenidone capsules, routine symptomatic treatments (e.g., oxygen therapy, antitussives, and antibiotics) were administered.

Bai (2019) [21] used the Maimendong decoction. No treatment was administered other than HM or pirfenidone. HM was not modified; only light meals, regular daily life, and maintaining calm emotions were encouraged.

Bian (2018) [22] used the Feixian decoction. Although modifications to herbal medicines were not permitted, antibiotics could be administered if necessary. Some lifestyle recommendations were provided, such as receiving health education, smoking cessation, avoiding dust/smoke/toxic gases, and light meals.

Liu (2017) [23] applied the modified Maimendong decoction. No additional customized modifications were made. Prednisone, a glucocorticoid, was prescribed in proportion to the weight of the patient. Symptomatic treatments and antibiotics were administered according to the condition of each patient. Lifestyle recommendations were also provided (e.g., smoking cessation, alcohol cessation, and low-flow oxygen inhalation).

*3.3. Risk of Bias (RoB)*

3.3.1. Random Sequence Generation

Five studies mentioned randomization. Random number tables were employed in three trials [18,20,23]. Two studies stated adopted complete randomization but did not describe any methods in detail [21,22]. We evaluated these five trials as having a low RoB. Li et al. [19] allocated participants according to the order of registration, leading to a high RoB.

3.3.2. Allocation Concealment

None of the six trials mentioned details on allocation concealment [18–23]. In other words, every study had an unclear RoB.

3.3.3. Blinding: Participants, Personnel, and Outcome Assessment

We evaluated the blinding criteria from two perspectives: blinding of participants and personnel and blinding of outcome assessors. Regarding the blinding of participants and personnel, all six trials were conducted completely non-blinded, naturally resulting in a high RoB [18–23]. Meanwhile, none of the studies made any records regarding the blinding of outcome assessors.

### 3.3.4. Incomplete Outcome Data Addressed

All six studies were examined to see whether they had incomplete outcome data, and there were no missing data or dropouts [18–23]. All studies had a low RoB.

### 3.3.5. Selective Reporting

None of the trials had a protocol published in advance, leading to a high RoB [18–23].

### 3.3.6. Other Bias

Other possible RoBs, such as baseline data not being comparable, were also assessed. No study showed a potential RoB (Figures 2 and 3) [18–23].

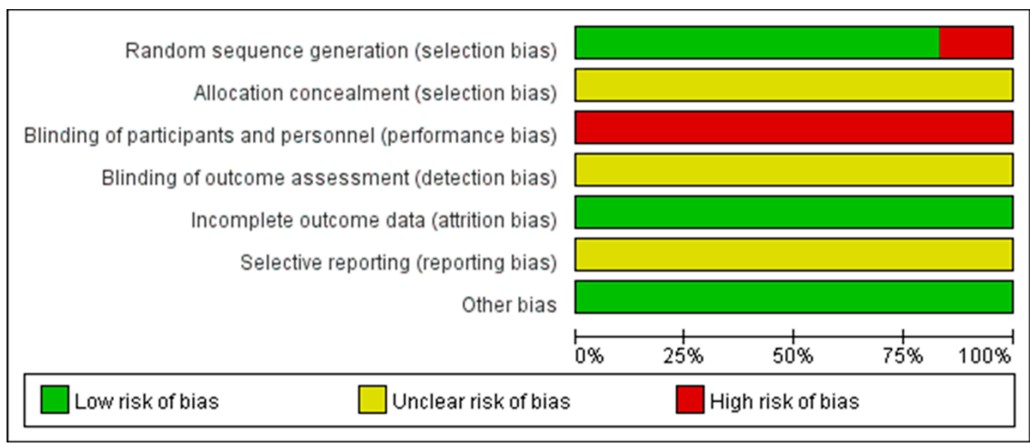

**Figure 2.** Risk of bias graph.

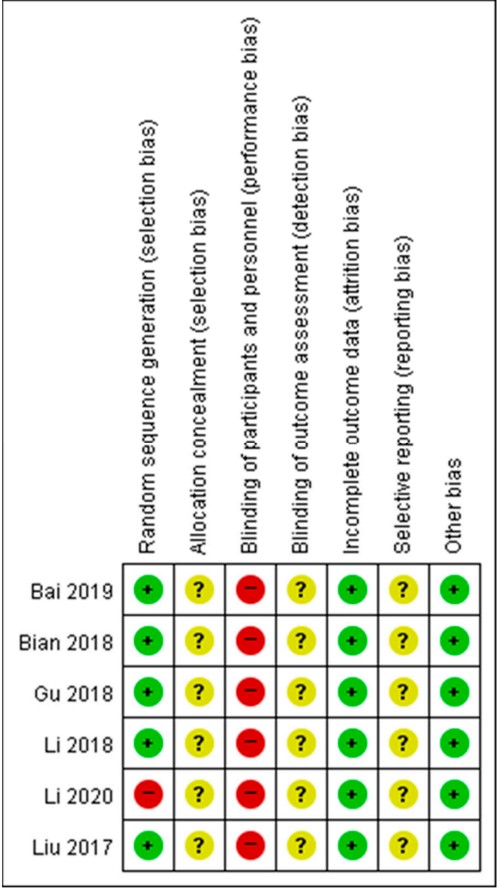

**Figure 3.** Summary of the risk of bias [18–23].

## 4. Discussion

This SR summarizes the efficacy and safety of herbal medicines administered along with pirfenidone in patients with IPF. We examined six RCTs that covered 569 participants. These studies commonly noted that pirfenidone could be more effective in treating IPF when combined with HM and, at the same time, could reduce the possibility of AEs. However, as mentioned in most studies, HM treatments for IPF are based primarily on clinical experience without sufficient evidence. Carefully designed studies are needed to provide more evidence.

The most common AEs covered in this review were gastrointestinal and skin-related symptoms. The key measure in this review was FVC, which reflects pulmonary function. IPF, the main disorder of this review, is a chronic, progressive respiratory disease with dry cough as the major symptom [1], and dyspnea and general weakening often occur as the disease progresses. From a traditional medical point of view, IPF is known to be mainly caused by Qi deficiency, lung Yin deficiency, and lung dryness due to Yin deficiency or lung heat, blood stasis, phlegm, and dampness [23,27]. In treating cough caused by lung dryness, moistening the lung has been regarded as a key therapeutic approach for the management of IPF symptoms. For this reason, the HM covered in this review commonly included herbs that can moisturize the lungs, replenish Qi, or supplement Yin. Antitussive herbs are commonly used to relieve symptoms. Lung moistening herbs (e.g., *Trichosanthis Semen*, *Fritillariae Cirrhosae Bulbus*, *Liriopis seu Ophiopogonis Tuber*, *Polygonati Rhizoma*) were used in every trial [18–23], and most of these are also effective in Yin supplements. Herbs to replenish Qi, such as *Codonopsis Pilosulae Radix* and *Astragali Radix,* were also used in high doses [20,22,23]. *Platycodonis Radix*, a representative antitussive herb, was also included in four trials [18,19,21,22].

For pirfenidone, the recommended dose by the pharmaceutical company was 801 mg three times a day. However, this can be adjusted for various reasons, such as reducing AE. The clinical dosage of pirfenidone varies [28]. According to real-world retrospective analysis, pirfenidone was administered at different concentrations of 400–1800 mg per day. Furthermore, low-dose administration of <1200 mg per day effectively prevents FVC reduction [29]. In the clinical studies covered in this review, the dose of pirfenidone ranged from 400 [18] to 1800 mg [22] per day. We speculated that the preventive effect of pirfenidone on the decline in FVC would have been effective.

In the included trials, the combination of pirfenidone and HM was more effective than pirfenidone alone. For studies that have defined "valid improvement" [18,22,23], the improvement rate was significantly higher in the HM-treated group. The remaining three trials also showed a remarkable improvement in the outcome measures, including FVC, compared to the control group [19–21]. In particular, all indicators of Li (2020) [19] and Bai (2019) [21] showed remarkable enhancement compared to the control. Furthermore, many secondary outcomes, such as serum cytokine levels and SF-36 scores, showed substantially better results in the treatment group [19–21].

There are some cases where the situation deteriorates (indicated by superscript[b] in Table 1); however, this does not mean that the interventions were ineffective. This is due to the irreversible and progressive nature of IPF. It is meaningful that it showed better results than the control group.

Regarding the safety assessment, three studies [19,22,23] mentioned minor AEs, such as rash and gastrointestinal discomfort. AEs occurred more frequently in the control group than in the HM group. This indicates that HM is safe and effective in reducing pirfenidone AEs. AEs caused by pirfenidone are common, and the most common AEs are gastrointestinal and skin-related symptoms. In addition, they often result in dose adjustments or treatment discontinuation in the real-world [16,28,30]. It would be clinically significant if HM administered with pirfenidone could contribute to dose maintenance by reducing AEs.

In terms of QoL, which was one of the limitations of pirfenidone, we were able to see the possibility that HM can work as a supplement. The overall improvement over that of

the control group was significant. However, as only three studies mentioned the impact of the SGRQ based on statistics, we need additional data to support this.

This SR was the first to examine the clinical efficacy of the parallel administration of herbal medicines in IPF compared to pirfenidone alone. Previous studies have examined the efficacy of herbal medicines in IPF. However, most of these studies were clinical trials. Pirfenidone was not specifically set as an intervention for SRs. Previous systematic reviews have compared the efficacy of herbal medicine to N-acetylcysteine, covering studies conducted from 2012 to 2018 [31], or of a single herb to all conventional treatments for IPF, covering studies conducted from 2005 to 2019 [17]. Considering that pirfenidone obtained FDA approval in 2014 and received recommendations only in the latest ATS/ERS/JRS/ALAT guidelines, previous SRs covering studies before 2014 have limitations. This is especially critical because the latest clinical practice guidelines of ATS/ERS/JRS/ALAT recommend against the use of N-acetylcysteine [9]. This study is expected to be of clinical utility because it only reviewed clinical trials in which pirfenidone was applied to the control and treatment groups.

We provisionally concluded that the administration of HM in combination with pirfenidone has advantages over pirfenidone monotherapy in terms of efficacy and safety. However, there are a few aspects to consider when reading this review.

First, it included only a small number of studies. As IPF is a rare disease, no previous studies have been conducted. Only six trials were available for this review. For this reason, meta-analysis and the GRADE (Grading of Recommendations Assessment, Development and Evaluation) assessment were not performed. To make up for this, we underwent qualitative synthesis, as can be seen in Table 1. Second, all trials were conducted and published in a single country in China. This may lack generality because the RCTs did not target diverse patients. Direct application to readers outside of China may have some limitations. Third, this review consisted of studies with a high or unknown RoB, and each RoB may have affected the results. All studies included in this systematic review had unclear or high RoB of allocation concealment, blinding of participants and personnel, blinding of outcome assessment, and selective reporting. Therefore, we look forward to well-designed, larger-scale RCTs being performed in the future. The fourth and final point is that some trials did not measure FVC, which is the key measure of IPF and the primary outcome of this review. Li (2018) [18] and Liu (2017) [23] used FEV1 rather than FVC. Although the data were supplemented with DLCO, as it is the gold standard for IPF, it would be ideal for future clinical trials to measure FVC.

## 5. Conclusions

In the treatment of IPF, the combined use of pirfenidone and herbal medicine was effective in protecting pulmonary function and improving QoL. However, the evidence was not strong enough to draw a definitive conclusion considering the number and quality of the included trials. Therefore, stricter RCTs are warranted to support the findings of this review.

**Supplementary Materials:** The following supporting information can be downloaded at: https://www.mdpi.com/article/10.3390/pr10122477/s1, Table S1: PRISMA 2020 Checklist; Table S2: Search strategy.

**Author Contributions:** Conceptualization, S.P., K.-I.K. and B.-J.L.; methodology, S.C., S.P. and J.A.L.; writing—original draft, S.C. and S.P.; writing—review and editing, K.-I.K., B.-J.L. and H.-J.J.; funding, B.-J.L.; supervision, K.-I.K. and B.-J.L. All authors have read and agreed to the published version of the manuscript.

**Funding:** This research was supported by a grant of the Korea Health Technology R&D Project through the Korea Health Industry Development Institute (KHIDI), funded by the Ministry of Health & Welfare, Republic of Korea (grant number: HI20C1205), (HI20C1405).

**Institutional Review Board Statement:** Ethical review and approval were waived for this study, as this study is a systematic review of previously published studies.



**Informed Consent Statement:** Patient consent was waived, as this study is a systematic review of previously published studies.

**Data Availability Statement:** The data presented in this study are available in the article and Supplementary Materials.

**Conflicts of Interest:** The authors declare no conflict of interest.

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
