# Peer review of "The Efficacy and Safety of Herbal Medicine with Pirfenidone in the Treatment of Idiopathic Pulmonary Fibrosis: A Systematic Review"

_processes, doi:10.3390/pr10122477_

Round 1

Reviewer 1 Report

I have only one comment, these are not keywords: (efficacy; systematic review), change them.

2-       

Author Response

Comment

I have only one comment, these are not keywords: (efficacy; systematic review), change them.

Author response

Thank you for your comment. We removed the keywords ‘efficacy; systematic review’ and added ‘effectiveness; safety’ instead, as follows:

“ Keywords: IPF; herbal medicine; pirfenidone; effectiveness; safety ”

(Please see page 1, red words)

Reviewer 2 Report

The paper is of great interest and relevant. It is well presented and written.

However, some issues need to be improved before being accepted for publication to encourage reading by readers.

1. It is interesting to include the keywords used to carry out the systematic review.

2. In page 3 of 21, lines 98 and 99, the term metanalysis is included, however the study is a systematic review.

3. In page 3 of 21, line 135, it is comment that the authors contacted the corresponding authors to verify the data in the studies with insufficient information. What information are they referring to?

4. In any systematic review, it is necessary to evaluate the quality and level of scientific evidence of the selected articles, as well as their degree of recommendation, in order to compare their results. What criteria have been followed?

5. In page 5 of 21 lines 172 and 173 there is an error and repetition.

6. In page 5 of 21, line 208 refer to the study of Li (2018) that does not provide information on statistical results, how have you verified the results obtained?

Author Response

Comment 1

 It is interesting to include the keywords used to carry out the systematic review.

Author response

Thank you for your comment. We removed the keywords ‘efficacy; systematic review’ and added ‘effectiveness; safety’ instead, as follows:

 “ Keywords: IPF; herbal medicine; pirfenidone; effectiveness; safety ”

(Please see page 1, red words)

Comment 2

In page 3 of 21, lines 98 and 99, the term metanalysis is included, however the study is a systematic review.

Author response

Thank you for your comment. We revised the sentence and meta-analysis has been deleted.

(Please see page 3, red words)

Comment 3

In page 3 of 21, line 135, it is comment that the authors contacted the corresponding authors to verify the data in the studies with insufficient information. What information are they referring to?

Author response

Thank you very much for your review and comments. We quoted the expression from the protocol, which was an idiomatic phrase. There was no insufficient data. We deleted and revised the sentence.

(Please see page 3, marked with “track changes” function)

Comment 4

In any systematic review, it is necessary to evaluate the quality and level of scientific evidence of the selected articles, as well as their degree of recommendation, in order to compare their results. What criteria have been followed?

Author response

Thank you for your insightful comment.

In order to assess the quality of evidence with the GRADE system, a meta-analysis must be performed. However, we judged this review to be not suitable for a meta-analysis. This is due to the insufficient number of studies included. Also, every study included was at some unclear or high risk of bias.

We added a description in the discussion regarding this issue.

(Please see page 19, marked with “track changes” function)

Comment 5

In page 5 of 21 lines 172 and 173 there is an error and repetition.

Author response

Thank you for your comment. We corrected an error and repetition.

(Please see page 5, red words)

Comment 6

In page 5 of 21, line 208 refer to the study of Li (2018) that does not provide information on statistical results, how have you verified the results obtained?

Author response

Thank you for the comment.

Like other studies included, Li (2018) also compared SGRQ based on statistical evidence, and both groups showed significant improvement after intervention. The improvement in the intervention group was also significant (p<0.05) compared to that of the control.

We corrected the error, and the related contents have also been corrected in Table 1 and Discussion.

(Please see the following:

Page 5, Line 205-207 in red words; Page 8, Li (2018), superscript in red words; Page 18 in red word)